# Academic Performance in Institutionalized and Noninstitutionalized Children: The Role of Cognitive Ability and Negative Lability

**DOI:** 10.3390/children10081405

**Published:** 2023-08-17

**Authors:** Mariana Sousa, Manuela Peixoto, Orlanda Cruz, Sara Cruz

**Affiliations:** 1The Psychology for Development Research Center, Lusíada University of Porto, 4100-346 Porto, Portugal; saracruz@por.ulusiada.pt; 2Centro for Psychology, Universty of Porto, 4200-135 Porto, Portugal; nelinha.peixoto@gmail.com; 3Faculty of Psychology and Sciences of Education, University of Porto, 4200-135 Porto, Portugal; orlanda@fpce.up.pt; 4Department of Psychology, School of Philosophy, Psychology & Language Sciences, University of Edinburgh, Edinburgh EH8 9YL, UK

**Keywords:** academic performance, cognitive ability, negative lability, institutionalized children

## Abstract

More research is needed to understand the factors that contribute to low academic achievement in institutionalized children. The aim of this study was to investigate the relationship between cognitive and emotion regulation skills and academic performance, by comparing institutionalized and noninstitutionalized Portuguese children. The sample comprised 94 participants (46 institutionalized (22 boys) and 48 noninstitutionalized (23 boys) children), aged between 6 and 10 years, matched for age and sex. We used Raven’s Colored Progressive Matrices (RCPM) to measure cognitive abilities. Emotional regulation and negative lability were assessed using the Emotion Regulation Questionnaire (ERC). Academic performance was assessed with the Competence Academic Scale (CAS) of the Portuguese version of the Social Skills Rating System—Teacher Form (SSRS-T). Institutionalized children exhibited poorer academic performance than their noninstitutionalized counterparts (effect size, *η*^2^ = 0.174). Cognitive ability (*β* = 0.28) and negative lability (*β* = −0.28) were significant predictors of academic performance. In addition to institutionalization, cognitive ability, and the challenges of managing negative emotions may contribute to the observed differences in academic performance. Interventions aimed at fostering cognitive and emotional competencies may play a protective role for institutionalized children facing academic and social difficulties.

## 1. Introduction

In Portugal, when children and adolescents experience family adversity and maltreatment, they are frequently placed in residential care. While residential care provides a more comprehensive and ecological assessment of the influence of institutionalization on children’s developmental adjustment, it is important to acknowledge the widely described negative effects on their development [1,2]. Among these, evidence pinpoints the detrimental impact that institutionalization may have on children’s academic performance [2,3].

Academic performance may be conceptualized as the quantity and/or quality of performed work by children at school, resulting from their learning process [4,5]. It encompasses the achievement of specific educational goals and the level of knowledge acquired in a particular subject [6]. Academic performance significantly influences both short- and long-term educational attainment, engagement, and work income [3,7], as well as physical and mental well-being [8,9]. The link between poor academic outcomes and socioemotional adjustment problems has been well documented, showing that poor academic performance is a risk factor for the development of psychopathology in children (e.g., anxiety, depression, substance abuse, or disruptive behavior) [10].

Differences in academic performance between institutionalized and noninstitutionalized children have been described [3,7,11,12,13,14]. Longitudinal studies show that educational dissimilarities between in-care children and their peers often begin early in development, increase over time, and persist [15]. In-care children are at greater risk of difficulties in literacy and numeracy, as well as school failure and dropout [3,16,17,18,19,20,21], which often lead to emotional and social problems, increasing the risk of psychopathology [22].

Although there is evidence that institutionalized children are more likely to experience academic difficulties than noninstitutionalized populations, research into the factors underlying the poor academic outcomes observed in institutionalized children is inconclusive. Furthermore, these factors have not been consistently addressed [23]. Systematic reviews of the educational outcomes of in-care children have mainly focused on understanding whether residential care provides an effective service for children and its impact on children’s development, including academic performance, which is one of the developmental markers [24,25,26]. This highlights the importance of considering other aspects that may be related to the academic performance of in-care children, such as cognitive and emotional dimensions. 

### 1.1. Cognitive Ability and Academic Performance

Cognitive ability is well correlated with the quality of learning [27,28,29,30] and academic performance [31]. This relationship is observed both in children living with their biological families and in institutions [32].

Children’s academic success depends on their cognitive ability to solve conceptual and abstract problems, as well as their critical thinking skills [33]. Children use cognitive flexibility to learn specific content, develop academic skills, and perform academic tasks [8,32,34]. It helps children adapt their responses to perform specific objectives and tasks [35], learn abstract concepts, and regulate their emotions and behaviors in the classroom, leading to better academic results [36]. 

The cognitive difficulties of in-care children are supported by the existing evidence [37,38,39]. In-care children tend to exhibit lower cognitive abilities than noninstitutionalized children, as they score lower on standardized tests of intelligence [40], which affects their academic performance [41].

### 1.2. Emotion Regulation Skills and Academic Performance

Emotion regulation involves children’s ability to manage extreme states of psychological arousal or reactivity, enabling them to respond more appropriately to the demands of the environment [42,43,44,45]. Although they are two distinct concepts, emotion regulation and negative lability are closely related and dynamically interact. Emotion regulation is the ability to monitor, modulate, and modify the duration, intensity, and nature of emotional responses and behavioral reactivity [42,46]. Negative lability refers to the readiness with which children respond to emotionally arousing stimuli and their difficulties in recovering from negative emotional responses [47]. It is associated with extreme states of psychological arousal or reactivity, intense emotional changes, mood lability, little or no cognitive–emotional flexibility in reading and processing emotional cues, and difficulties in regulating negative affectivity [43,48]. 

To ensure learning effectiveness and academic success, children need to manage their emotions and redirect or divert them from complex cognitive functions, such as attention, planning skills, abstract reasoning, and working memory, which are key to the learning process [49]. Emotion regulation helps children to focus their attention on school tasks, and to process, remember, and/or retain information [50,51,52]. Children may use various strategies to manage their emotions while in the classroom: (i) cognitive re-evaluation (i.e., modifying the evaluation of the situation to influence its emotional impact); (ii) thinking (i.e., focusing attention on one’s own feelings); (iii) emotional expression suppression (i.e., inhibiting the expressive course of an emotion); (iv) distraction (i.e., shifting the focus of attention from the situation to other events) [42,49,53]; and (v) emotional management (i.e., changing the perceived ability to cope with the emotional cues associated with specific situations). Other emotion regulation strategies have also been described in terms of their impact on the learning process: (i) emotion recognition (i.e., appropriately associating semantic categories with emotional experiences) [54]; (ii) tolerance to distraction (i.e., persisting in performing an activity while displacing the experienced emotions) [55]; and (iii) modification of the valence, intensity, and/or the length of emotional responses [54].

Children who regulate their emotions effectively tend to successfully manage the potential negative effects of emotions in unfavorable situations and to use positive emotions to increase their willingness to learn and perform schoolwork appropriately [56]. This leads their teachers to rate their academic performance positively [57]. In contrast, children with poor regulation skills tend to have difficulty organizing their studies and sustaining attention when confronted with teachers’ instructions [58].

In addition, children who have difficulty regulating their emotions generally show more attention problems, as well as hyperactive and disruptive behavior in school [59,60,61]. Because children with greater emotional lability are often in a negative mood and have more difficulty dealing with frustration, they may have more difficulty concentrating on learning tasks and completing school assignments [62]. In summary, emotion regulation difficulties, closely associated with higher negative lability, tend to negatively affect children’s academic performance and increase the risk of school dropout [58].

Institutionalized children who are exposed to early adversity tend to exhibit greater emotion regulation difficulties than nonexposed children, irrespective of age, gender, or cultural differences [43,63,64,65,66,67,68]. Research shows that these children commonly use maladaptive regulation strategies (e.g., disengagement, rumination, and expressive suppression) and are less likely to use effective strategies (e.g., cognitive reappraisal) compared with children who have not experienced adversity [66,69,70,71].

### 1.3. Objectives and Hypotheses

Given the aforementioned evidence, it is imperative to delve into the factors underlying academic performance, especially in more vulnerable populations, such as children living in institutions, who often encounter significant learning difficulties and school failure. The investigation into the factors underlying poor academic performance remains inconclusive, requiring further in-depth and consistent analysis [23]. Here, we explored the effects of living in residential care, cognitive ability, emotion regulation, and negative lability on academic performance. Specifically, the aim was to examine differences in these dimensions in institutionalized and noninstitutionalized children and determine the potential predictive value of cognitive ability, emotion regulation, and negative lability on academic performance while controlling for institutionalization status. We hypothesize the following statements: (1) living in an institution negatively predicts academic performance; (2) cognitive ability positively predicts academic performance; (3) emotion regulation positively predicts academic performance; and (4) negative lability negatively predicts academic performance. In addition, it is expected that cognitive ability, emotion regulation, and negative lability predict academic performance when controlling for institutionalization status.

## 2. Materials and Methods

### 2.1. Participants

The sample comprised 94 participants, 46 institutionalized (22 boys; 47.83%) and 48 noninstitutionalized (23 boys; 47.92%) children, matched for sex and age (institutionalized: M = 8.20, SD = 1.0; noninstitutionalized: M = 8.40, SD = 1.0), which has been described in a previous article (cf., [72]). 

The institutionalized children enrolled in this study had experienced abuse or neglect, leading to their separation from their biological families. They were subsequently relocated to residential care facilities, following a court order and referral from Portuguese Child Protective Services. The selection of this group was based on a 2012 list from the Ministry of Social Affairs of the Portuguese residential care institutions in Viana do Castelo, Braga, Porto, and Aveiro districts. For participant selection, developmental delays, as identified by the institutional caregivers and teachers, were considered as exclusion criteria. Equally, children diagnosed with cognitive deficits by mental health professionals were not included in the study. Noninstitutionalized children lived with their biological and nuclear families (e.g., parents and siblings), and they were not exposed to any type of maltreatment. 

All children enrolled in public schools in the referred districts.

### 2.2. Instruments

#### 2.2.1. Sociodemographic Information

Data on the process of institutionalization process, interactions with the biological family, social support network, and medical history of institutionalized children were collected with the assistance of the caregivers from the respective institutions. Meanwhile, for children living with their biological families, data on family risk factors (e.g., drug and alcohol abuse, parents or substitutes psychopathology, and overt reactions defined as ‘others’), parental educational background, and medical history was collected in collaboration with elementary school teachers. The information on family risk factors allowed for the exclusion of children whose teachers identified any of the risk factors mentioned during data collection.

#### 2.2.2. Cognitive Ability

The Portuguese version (translated, validated, and adapted by Simões in 1995 [73]) of Raven’s Colored Progressive Matrices (RCPM) [74] was employed to assess children’s cognitive ability. It measures nonverbal intellectual performance in children aged 5 to 11 years and provides an overall score that reflects children’s level of intelligence [75,76]. The RCPM comprises 36 nonverbal items, divided into three series of 12 items: A, Ab, and B. During the assessment, children were presented with an image and asked to choose the correct completion option from six possibilities. The obtain the total score, the individual scores of the three series are summed up (maximum score of 36 points). Each series of the RCPM targets different intellectual dimensions: RCPM-A evaluates the ability to identify sameness, RCPM-Ab assesses visuoperceptual competencies, and RCPM-B measures abstract and analogical reasoning [77]. The consistency results for the RCMP series are good (RCMP-A α = 0.82; RCMP-Ab α = 0.87; and RCMP-B α = 0.87). 

#### 2.2.3. Emotion Regulation and Negative Lability

Emotion regulation and negative lability were measured with the Portuguese version of the Emotion Regulation Checklist (ERC) [78].

The ERC focuses on assessing children’s ability to cope with their emotional experiences. The checklist includes 24 items that are rated on a four-point Likert scale (1 = *never* to 4 = *almost always*). It is composed of two different subscales: Emotion Regulation (8 items) and Emotion Lability/Negativity (i.e., negative lability) (16 items). The items on the Emotion Regulation subscale measure a child’s ability to adaptively regulate their emotions, including aspects such as appropriate emotional responses in social situations, showing empathy, equanimity, knowledge, and self-awareness of their emotions. For example, it assesses how well a child can effectively control arousal or cope with frustration when faced with emotionally arousing situations. On the other hand, the Negativity Lability/Negativity subscale assesses arousal, responsiveness, flexibility, intensity of emotional responses, expression of negative affect, and mood lability. For example, it allows us to measure how quickly a child’s emotional response can shift from a positive to a negative mood, making it difficult to predict their reactions.

To obtain a total score, the sum of the items of each subscale is calculated. Higher scores on the Emotion Regulation subscale indicate that children can effectively modulate emotional arousal and respond adaptively to social situations. Conversely, higher scores on the Emotion Lability/Negativity subscale indicate a tendency toward rapid and extremely intense emotional reactions and mood swings. The construct and discriminant validity of the ERC have been previously documented (Shields and Cicchetti, 1997 [78]). Here, Cronbach’s alpha was acceptable for the Emotion Regulation subscale (α = 0.71) and good for the Emotion Lability/Negativity subscale (α = 0.88).

### 2.3. Academic Performance 

The children’s academic performance was assessed using the Competence Academic Scale (CAS) of the Portuguese version of the Social Skills Rating System—Teacher Form (SSRS-T) [79,80]. The SSRS was designed to assess social behavior in children and adolescents between the ages of 3 and 18 years. It consists of three scales: Social Skills, Problem Behavior, and Competence Academic scales. For the purposes of this study, only the CAS was used. This scale is composed of six items, rated on a five-point Likert scale (1 = *worse than average* to 5 = *better than average*). These items cover different aspects of academic and intellectual performance, including performance in reading/Portuguese language and mathematics skills, both in relation to their peers. It also takes into account reading/Portuguese language and mathematical skills in relation to what is expected at their school level.

The scores from the six items are added together to calculate the overall Academic Performance score. In this study, the reliability of this scale was found to be excellent (α = 0.97). 

### 2.4. Procedure

This study was ethically approved by the ethics committee of the institution where it was conducted and was carried out in accordance with the tenets of the Declaration of Helsinki. The aims and methodology were presented to the institutions and school directors via telephone, and they were invited to participate. After obtaining their consent, an e-mail was sent to those who agreed to be part of the study. With the assistance of technical staff from the institutions and school staff, the parents of the children were then contacted to give informed consent. Parents who gave permission for their children to participate also gave permission for their teachers and institutional caregivers to complete the questionnaires.

The questionnaires for institutionalized children were completed by their caregivers, while for noninstitutionalized children, the responsibility for completing them fell on their teachers. This happened because, in some cases, the families of institutionalized children were unavailable and contact with the children’s teachers depended on the availability of the institutional caregivers, which was not always possible.

Initially (time 1 occurred in 2012), the RCPM was administered to the children, a process that took approximately 15 to 20 min. The administration was performed by a group of clinical and educational psychologists experienced in clinical psychological assessment with children. A warm-up session was conducted before the time of assessment to establish a safe and positive interaction with them.

One year after the initial assessment (i.e., in 2013), the institutions and schools were contacted again to ask the caregivers and teachers to complete the ERC and CAS questionnaires. These took approximately five to ten minutes to complete. Caregivers and teachers completed the questionnaires independently and were given the opportunity to contact the first author if they had any doubts or needed clarification about their completion. The flowchart of the process is shown on Figure 1.

### 2.5. Data Analysis

A prior analysis was performed using G*Power version 3.1.9.6 to determine the sample size required for the study. To detect a medium effect size (f = 0.25) with 95% power in a two-group multivariate analysis of variance, a minimum sample size of 74 participants was determined. For hierarchical regression analysis with five predictors, a minimum sample size of 96 participants was required to detect a medium effect size (f = 0.25) with 85% power.

Statistical procedures and analyses were performed using IBM SPSS software version 26.0. Descriptive statistics, including means, standard deviations, ranges, and frequencies, were calculated to characterize the sample. A multivariate analysis of variance was then performed to assess the differences between all the variables studied according to the children’s institutionalization status (institutionalized vs. noninstitutionalized). Finally, a hierarchical regression analysis was carried out to explore the predictors of academic performance. In step 1 of the regression, institutionalization status (institutionalized vs. noninstitutionalized) was included as a predictor. In step 2, cognitive ability and emotion regulation skills (emotion regulation and negative lability) were also included as additional predictors.

## 3. Results

### 3.1. Cognitive Ability, Emotional Regulation Skills, and Academic Performance

Table 1 shows the means, standard deviations, and ranges of responses for academic performance, cognitive ability, and emotion regulation (emotion regulation and negative lability). A multivariate analysis of variance was conducted to examine the differences in these variables between institutionalized and noninstitutionalized children.

The results indicated significant main effects for group (*F*(4, 85) = 5.05, *p* = 0.001, *η*^2^ = 0.192). The univariate analyses are detailed in Table 1. Significant differences were observed for cognitive ability (*F*(1, 88) = 18.58, *p* < 0.001, *η*^2^ = 0.174), with noninstitutionalized children scoring significantly higher (95% CI 21.59–24.67) than institutionalized children (95% CI 16.70–19.91). No significant differences were found for academic performance (*F*(1, 88) = 1.22, *p* = 0.273, *η*^2^ = 0.014); emotion regulation (*F*(1, 88) = 0.01, *p* = 0.947, *η*^2^ < 0.001); or negative lability (*F*(1, 88) = 2.51, *p* = 0.117, *η*^2^ = 0.028).

### 3.2. Cognitive Ability and Emotion Regulation Skills as Predictors of Academic Performance

To examine the predictive value of cognitive ability and emotion regulation (emotional regulation and negative lability) above and beyond the institutionalization status, a hierarchical regression analysis was conducted (see Table 2). In the first step of the hierarchical regression, institutionalization status was included as a predictor. It was found that institutionalization status did not significantly predict academic performance (*β* = 0.12, *p* = 0.273). In the second step, cognitive ability, emotion regulation, and negative lability were added as additional predictors. After controlling for institutionalization status (*β* = −0.05, *p* = 0.671), the combined variables accounted for 20% of the variance in academic performance. In the analysis, cognitive ability (*β* = 0.28, *p* = 0.011) emerged as a significant positive predictor of academic performance, whereas negative lability (*β* = −0.28, *p* = 0.009) emerged as a significant negative predictor of academic performance.

## 4. Discussion

The existing literature has extensively highlighted the negative impact of institutionalization on the academic performance of in-care children, often leading to severe learning difficulties and school failure [16,18,19,20,81]. However, research into the underlying factors affecting this vulnerable population remains scarce and inconsistent [23]. A comprehensive and detailed analysis was needed to fill this research gap. To address this need, this study examined the effects of institutionalization, cognitive ability, emotion regulation, and negative lability on academic performance. The main goals were twofold: (1) to analyze and compare the differences in academic performance, cognitive ability, emotion regulation, and negative lability, between institutionalized and noninstitutionalized children; and (2) to assess the relative importance of institutionalization, cognitive ability, emotion regulation, and negative lability as predictors of academic performance.

In terms of group (i.e., institutionalized and noninstitutionalized children) differences in academic performance, cognitive ability, emotion regulation, and negative lability, the results revealed that institutionalized children exhibited lower cognitive performance than noninstitutionalized children. These findings align with existing research, which has shown that in-care children often experience cognitive difficulties [37,38,39,68]. In-care children tend to score lower on standardized tests of cognitive performance [40], and these cognitive difficulties negatively impact their academic performance and overall adjustment [41].

In this study, no statistically significant differences in academic performance, emotion regulation, and negative lability were found between institutionalized and noninstitutionalized children. The lack of differences in academic performance could be attributed to the resilient coping strategies that institutionalized children may develop to cope with the challenges they face at school. This is coherent with studies reporting that institutionalization may protect children from developing emotional and behavioral problems [82,83,84]. It is possible that living in an institution may help children to establish structured routines, especially in relation to their schoolwork, which may have a positive impact on their learning process and academic performance. In addition, the institutional environment allows these children to develop supportive and nurturing relationships with adults and peers, which helps them to cope effectively with academic and social difficulties. Furthermore, institutionalized children often benefit from interventions that focus on improving their social and emotional skills, which contributes to the development of adaptive emotional regulation strategies [25,85]. Comparing the relative importance of institutionalization, cognitive ability, emotion regulation, and negative lability in predicting academic performance, it was found that cognitive ability and negative lability predicted academic performance, whereas institutionalization and emotion regulation showed no predictive value. These findings suggest that in addition to institutionalization, poor cognitive ability, such as flexibility or abstract reasoning difficulties, and negative lability may act as risk factors for institutionalized children. 

Cognitive difficulties in institutionalized children have been well described [37,38], as has their negative impact on these children’s academic performance [41]. Children’s educational achievement depends on their ability to use their cognitive skills to solve conceptual and abstract problems, as well as their critical thinking [33], and cognitive flexibility [8,32,34]. As a result, in-care children often experience substantial difficulties in coping with school tasks and learning [40], as their cognitive difficulties often interfere with their ability to grasp abstract concepts and effectively regulate their emotions and behaviors in the classroom [36].

The association between difficulties in emotion regulation and academic performance and engagement in school is also supported by existing research [58,86,87]. Children with difficulties in regulating their emotions tend to experience difficulties in organizing their studies and show more problems in responding adaptively to classroom demands, such as following teacher instructions [58]. Furthermore, difficulties in emotion regulation are strongly associated with attention problems and hyperactivity [59,60,61], which also have a negative impact on academic performance. Children with higher levels of negative lability typically express intense changes in their emotional states, poor flexibility, severe mood swings, and difficulty managing negative affect [43,48]. They are often in a negative mood and have difficulty coping with frustration, which can lead to difficulties in focusing their attention on school tasks [62].

Negative lability is associated with poor academic performance [58]. These difficulties are exacerbated in institutionalized children, whose inappropriate emotion regulation strategies have already been reported in other studies [43,63,64,66,67,88]. It has been shown that in-care children are more likely to use maladaptive regulation strategies (e.g., disengagement, expressive suppression, and rumination) and less likely to use effective strategies (e.g., cognitive reappraisal) than children who have not been exposed to adversity [66,69,70,71]. In addition, because abusive parents regularly have difficulty helping their children manage negative emotions or use maladaptive strategies such as dismissing children’s emotions and/or humiliating them, they tend to provide fewer models of adaptive emotion regulation [89,90]. Subsequently, the impaired self-regulation skills observed in in-care children inevitably have a detrimental effect on their academic performance, learning, and educational achievement.

### Limitations and Future Directions

There are several limitations to the present study. The first limitation is the reduced number of children in each group. Other studies should include larger sample sizes to corroborate the findings reported here. Also, the RCPM test may prove easier for older children than for younger children, due to the maturation of cognitive structures that support abstract reasoning. Another limitation to be addressed is the use of different informants for each group, which may introduce potential bias or variability in the data collection process. Future research should consider using the same informants for institutionalized and noninstitutionalized children. A multi-informant assessment should also be included to allow for a comprehensive and ecological analysis of children’s development and adjustment, as well as to compare the perceptions of parents/substitutes with the perceptions of teachers regarding children’s functioning. Finally, the effect of parental level of education was not controlled for in the noninstitutionalized group. Future research should address this effect.

## 5. Conclusions

This study offers insight into the factors underlying poor academic performance among institutionalized and noninstitutionalized children. The results show that cognitive ability and difficulties in modulating negative emotions may help explain the observed differences in academic performance more than institutionalization. Thereafter, interventions targeting cognitive and emotional competencies in institutionalized children are necessary, as they appear to act as protective factors in coping with academic and social demands. In addition, institutions and agencies that work closely with at-risk children may benefit from psychoeducational programs and structured interventions to promote children’s emotion regulation skills, focusing on the management of negative emotions. They may also benefit from educational and psychological interventions to address and promote neurocognitive dimensions related to cognitive performance. In summary, psychoeducational interventions targeting emotion regulation skills and cognitive ability may help improve the well-being and psychopathological adjustment of institutionalized children.

## Figures and Tables

**Figure 1 children-10-01405-f001:**
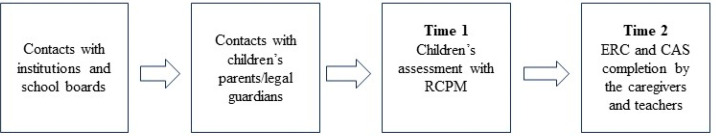
Flowchart of the participants’ recruitment and data collection process.

**Table 1 children-10-01405-t001:** The means, standard deviations, and ranges of responses regarding academic performance, cognitive ability, and emotional skills for the global sample (*N* = 94), and for institutionalized (*n* = 46) and noninstitutionalized children (*n* = 48), separately, and between-group differences.

	Total (*N* = 94)		Institutionalized (*n* = 46)		Noninstitutionalized (*n* = 48)		Between-Group Differences		
	M (SD)	Range	M (SD)	Range	M (SD)	Range	*F*(1, 88)	*p*	*η* ^2^
Academic performance	2.85 (0.81)	1.00–5.00	2.75 (0.78)	1.00–5.00	2.94 (0.84)	1.00–5.00	1.22	0.273	0.014
Cognitive ability	20.82 (5.81)	9.00–34.00	18.30 (5.29)	9.00–28.00	23.13 (5.32)	10.00–34.00	18.58	<0.001	0.174
Emotion regulation	23.11 (3.82)	14.00–49.00	23.14 (3.71)	14.00–42.00	23.09 (3.96)	15.00–49.00	0.01	0.947	<0.001
Negative lability	28.23 (7.38)	14.00–31.00	29.51 (6.91)	14.00–31.00	27.06 (7.68)	17.00–31.00	2.51	0.117	0.028

**Table 2 children-10-01405-t002:** Hierarchical regression for cognitive ability and emotional skills as predictors of academic performance, controlling for institutionalization status (*N* = 94).

	*β*	95% CI	*t*	R^2^	ΔR^2^
Step 1				0.014	0.014
Institutionalization status	0.12	−0.15–0.53	1.10		
Step 2				0.200	0.186 ***
Institutionalization status	−0.05	−0.42–0.27	−0.43		
Cognitive ability	0.28 *	0.01–0.07	2.60		
Emotion regulation	0.11	−0.02–0.07	1.04		
Negative lability	−0.28 **	−0.06–−0.01	−2.66		

Note: Institutionalization status: being institutionalized vs. not being institutionalized; * *p* < 0.05; ** *p* < 0.01; *** *p* < 0.001; *β*—standardized effect.

## Data Availability

The data that support the findings of this study are available upon request to the corresponding author. The data are not publicly available due to privacy or ethical restrictions.

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
