# Peer review of "Academic Performance in Institutionalized and Noninstitutionalized Children: The Role of Cognitive Ability and Negative Lability"

_children, 2023, doi:10.3390/children10081405_

Round 1
Reviewer 1 Report
Manuscript entitled: Academic performance in institutionalized and noninstitutionalized children: The role of cognitive ability and negative lability.
I really appreciated the authors’ hard work for keeping this study happened. However, there are some issues to be concerned about and I need the authors to clarify as followed.
1. It would be better if the authors add the license number for the IBM SPSS version 26.0 they used in statistical analysis.
2. The authors mentioned that the institutionalized children were removed from their biological families and placed in residential care, after being victims of abuse or negligence, due to Court order and after being referred by the Portuguese Child Protective Services. Could authors specify the year in which the Court order? Specifying the year of the Court order will better understand the behavioral changes of the participants after the study.
3. The selection of this group was based on a list of the Ministry of Social Affairs of the Portuguese residential care institutions in these cities. Could authors specify the year of the Ministry of Social Affair of the Portuguese residential care institutions? Also, the authors mentioned “…. in these cities”, what are cities?
4. The author mentioned that developmental delays identified by the institutional caregivers and teachers were considered as exclusion criteria. How? Please kindly describe!
5. The authors mentioned that all children attended public schools in Northern Portugal. Did all children from both groups? Or just only the institutional group?
6. The authors mentioned that the elementary school teachers and the main caregivers in the institutions, respectively for the noninstitutionalized and institutionalized groups, were invited and asked to fill out a sheet to collect information on the children and families’ socio-demographics (i.e., institutionalization process, contacts with the biological family, social support network, and children’s medical history). Did the family members also give the information, especially those who are in the non-institutional group?
7. A flow chart of methodology might be needed for this manuscript as it would help the readers to understand the whole process of data collection and the outcomes.
8. The authors mentioned that this study was reviewed and approved by the Ethics Committee of the institution where it was conducted. The Ethical Committee Approval Number and date of protocol must be mentioned.
9. When was this study conducted?
10. Finally, almost 60% must be later than 5-6 years. The current version of the manuscript cited those references later than 10 years. The citation must be improved.
After the authors could declare all the above concerns, I would be happy to reconsider the revised manuscript.
Author Response
Reviewer 1
I really appreciated the authors’ hard work for keeping this study happened. However, there are some issues to be concerned about and I need the authors to clarify as followed.
Authors’ response: We appreciate and acknowledge the reviewer’s comments. We are also grateful for all the effort put into the analysis of the manuscript and for all the helpful comments and suggestions provided.
1. It would be better if the authors add the license number for the IBM SPSS version 26.0 they used in statistical analysis.
Authors’ response: Thank you for the suggestion. It is not common to include the license number in the manuscript. Nevertheless, the software used in this study was provided by Lusíada University – Porto, under an appropriate license.
2. The authors mentioned that the institutionalized children were removed from their biological families and placed in residential care, after being victims of abuse or negligence, due to Court order and after being referred by the Portuguese Child Protective Services.Could authors specify the year in which the Court order? Specifying the year of the Court order will better understand the behavioral changes of the participants after the study.
Authors’ response: Thank you for the careful analysis and insightful comment. We acknowledge that temporarily framing the Court order regarding the removal of the children from their biological families would be beneficial. However, unfortunately, we did not have access to this information.
3. The selection of this group was based on a list of the Ministry of Social Affairs of the Portuguese residential care institutions in these cities.Could authors specify the year of the Ministry of Social Affair of the Portuguese residential care institutions?Also, the authors mentioned “…. in these cities”, what are cities?
Authors’ response: Thank you for the useful suggestions. Information on the year in which the list was consulted and the location of the institutions has been added (cf., lines 160-162).
4. The author mentioned that developmental delays identified by the institutional caregivers and teachers were considered as exclusion criteria.How? Please kindly describe!
Authors’ response: Thank you for the detailed analysis of the manuscript. The required information was described in the text.
5. The authors mentioned that all children attended public schools in Northern Portugal.Did all children from both groups? Or just only the institutional group?
Authors’ response: Thank you for the comment. Children from both groups attended public schools, as stated in line 188 of the first version of the manuscript (i.e., ‘All children attended public schools in Northern Portugal.’).
6. The authors mentioned that the elementary school teachers and the main caregivers in the institutions, respectively for the noninstitutionalized and institutionalized groups, were invited and asked to fill out a sheet to collect information on the children and families’socio-demographics (i.e., institutionalization process, contacts with the biological family, social support network, and children’s medical history).Did the family members also give the information, especially those who are in the non-institutional group?
Authors’ response: Thank you for the question. The family members were not engaged in data collection. Only the elementary teachers and the main caregivers from the institution were asked to provide the requested information.
7. A flow chart of methodology might be needed for this manuscript as it would help the readers to understand the whole process of data collection and the outcomes.
Authors’ response: Thank you for the useful suggestion. A flow chart was added to illustrate the procedures regarding participants’ selection and data collection (cf., lines 259-260).
8. The authors mentioned that this study was reviewed and approved by the Ethics Committee of the institution where it was conducted.The Ethical Committee Approval Number and date of protocol must be mentioned.
Authors’ response: Thank you for bringing up this relevant issue. While we did include the date of the protocol (cf., Institutional Review Board Statement), there is no associated protocol number. This document was sent to the Editorial Assistant, after the manuscript has been submitted.
9. When was this study conducted?
Authors’ response: Thank you. This study was performed between 2012 and 2013.
10. Finally, almost 60% must be later than 5-6 years. The current version of the manuscript cited those references later than 10 years. The citation must be improved.
Authors’ response: Thank you for pointing this out. More recent references were added.
After the authors could declare all the above concerns, I would be happy to reconsider the revised manuscript.
Reviewer 2 Report
I don't think this is a qualified manuscript. First of all, the authors have not fully discussed the theoretical basis and research assumptions, and the relationship between variables is very weak. Secondly, the analysis of research results. Obviously, the existing results reports are not sufficient, and they lack many necessary results reports. Finally, about the conclusion, the part of the conclusion is not effectively combined with the previous results. At the same time, there are many problems in grammar and format.
Author Response
I don't think this is a qualified manuscript. First of all, the authors have not fully discussed the theoretical basis and research assumptions, and the relationship between variables is very weak. Secondly, the analysis of research results. Obviously, the existing results reports are not sufficient, and they lack many necessary results reports. Finally, about the conclusion, the part of the conclusion is not effectively combined with the previous results. At the same time, there are many problems in grammar and format.
Authors’ response: Thank you for the time and effort you have invested in analyzing the manuscript. We acknowledge that the manuscript may have some weaknesses, and we have made substantial improvements by incorporating more recent references, clarifying methodological issues, and revising the grammar, in line with the suggestions provided by the Editor and Reviewer 1.
Round 2
Reviewer 1 Report
Q.ver.1.: 2. The authors mentioned that the institutionalized children were removed from their biological families and placed in residential care, after being victims of abuse or negligence, due to Court order and after being referred by the Portuguese Child Protective Services. Could authors specify the year in which the Court order? Specifying the year of the Court order will better understand the behavioral changes of the participants after the study.
A.ver.1: 2. Authors’ response: Thank you for the careful analysis and insightful comment. We acknowledge that temporarily framing the Court order regarding the removal of the children from their biological families would be beneficial. However, unfortunately, we did not have access to this information.
Suggestion: However, if the authors could not specify the year of the Court order, the authors can mention the period of the Court order instead because it would help the data collection and procedure become strong procedure.
Q.ver.1.: 9. When was this study conducted?
A.ver.1: 9. Authors’ response: Thank you. This study was performed between 2012 and 2013.
Suggestion: In the manuscript, the authors should mention that this study was performed between 2012 and 2013.
Author Response
Q.ver.1.: 2. The authors mentioned that the institutionalized children were removed from their biological families and placed in residential care, after being victims of abuse or negligence, due to Court order and after being referred by the Portuguese Child Protective Services. Could authors specify the year in which the Court order? Specifying the year of the Court order will better understand the behavioral changes of the participants after the study.
A.ver.1: 2. Authors’ response: Thank you for the careful analysis and insightful comment. We acknowledge that temporarily framing the Court order regarding the removal of the children from their biological families would be beneficial. However, unfortunately, we did not have access to this information.
Suggestion: However, if the authors could not specify the year of the Court order, the authors can mention the period of the Court order instead because it would help the data collection and procedure become strong procedure.
Authors’ response: Thank you for the attentive and careful analysis of the revised version of the manuscript. We agree with the reviewer that knowing the period of the Court order would be extremely informative and we acknowledge the importance of considering the children’s length of institutionalization. However, the placement of the children in residential care varied greatly and depended on each child's circumstances. Some children were placed in residential care for longer periods, lasting over a year, while others were in care for only a month or two. As a result, unfortunately, it is not possible to pinpoint a specific date for the decree of the promotion and protection measure of Residential Care, as it depended on the individual process of each child.
Q.ver.1.: 9. When was this study conducted?
A.ver.1: 9. Authors’ response: Thank you. This study was performed between 2012 and 2013.
Suggestion: In the manuscript, the authors should mention that this study was performed between 2012 and 2013.
Authors’ response: Thank you. Information concerning the time when data collection occurred was added to the manuscript (line 249 and line 254).

Reviewer 2 Report
I think there are still many shortcomings in this manuscript, please make some revision about the grammar and references. I hope that the authors can make meaningful and practical research in future research.
Author Response
We are grateful for the analysis of the revised version of the manuscript. We included some additional changes in the manuscript that we believe to have improved its quality, namely in what concerns grammar and the references.